# A Broadband Vortex Beam Generator Based on Single-Layer Hybrid Phase-Turning Metasurface

**DOI:** 10.3390/mi14020465

**Published:** 2023-02-17

**Authors:** Cheng Fu, Jianing Zhao, Fang Li, Hao Li

**Affiliations:** 1College of Information Science and Engineering, Guilin University of Technology, Guilin 541006, China; 2Guangxi Key Laboratory of Embedded Technology and Intelligent System, Guilin University of Technology, Guilin 541004, China; 3School of Electronic Science and Engineering, University of Electronic Science and Technology of China, Chengdu 610054, China; 4Yangtze Delta Region Institute (Huzhou), University of Electronic Science and Technology of China, Huzhou 313001, China

**Keywords:** metasurface, wideband, orbital angular momentum (OAM), Ka-band

## Abstract

Vortex beams carrying orbital angular momentum (OAM) have become a research frontier due to the prospect of improving spectral efficiency and transmission capacity in communication systems. In this work, a hybrid phase-turning meta-atom that combines resonance and geometric (Pancharatnam-Berry) phase modulation is used to form a single-layer metasurface. A linearly polarized broadband vortex beam of mode *l* = −1 is obtained by the metasurface. An experimental prototype of the vortex beam generator has been fabricated and measured. The simulated and measured results demonstrate that the whole vortex beam generator exhibits over 70% mode purity from 26.5 GHz to 40 GHz (the relative bandwidth is 38.57%). In addition, a wide 3 dB gain bandwidth and low crosstalk are also provided by the proposed generator. This indicates that the proposed generator has important application value for vortex beam communication and its related applications.

## 1. Introduction

Vortex beams carrying orbital angular momentum (OAM) have gained a lot of interest due to great development prospects in communication capacity and spectral efficiency [1,2,3,4,5]. The generation of vortex beams has been explored in many different ways by researchers, such as circular arrays [6,7], active diffraction gratings [8], spiral phase plates [9], and spiral paraboloids [10]. However, these approaches have the problems of complex feeding networks or large sizes. As a subwavelength artificial periodic metamaterial, metasurfaces offer great advantages such as strong regulation ability, simple feeding structure, low cost, and low profile, which have been employed in hologram imaging [11,12], wavefront phase control [13], flat lenses [14,15], absorbers [16,17], etc.

The initial study of the vortex beams based on metasurfaces mainly focused on generation methods [18,19,20]. In order to quantify the quality of the generated vortex beam, the mode purity and mode purity bandwidth of OAM have progressively emerged as the primary issue of concern. On the one hand, higher purity vortex beams can not only effectively reduce mode dispersion and crosstalk, but also facilitate receiver detection and identification [21]. On the other hand, a wide bandwidth in vortex beams can reduce the distortion caused by chromatic dispersion as well as develop some new applications [22,23,24]. Hence, there are some recent studies and reports on broadband vortex beam generators [25,26,27,28,29,30]. For instance, Guo et al. design a reconfigurable metasurface with the octagonal ring slot structure, which aims to produce vortex beams with multiple different OAM modes in a broad operation band. The measured mode purity is more than 60%, and the mode purity bandwidth is about 11% [25]. Li et al. demonstrate an OAM reflective metasurface with half-wavelength rectangular dielectric elements made with 3D-printed technology. Experimental results show that the metasurface obtains a mode purity of 93.8% at 30 GHz. When mode purity exceeds 60%, the mode purity bandwidth is only 21.7% [26]. Moreover, Wu et al. construct a metasurface through a four-layer dielectric substrate. The above design achieves a mode purity bandwidth of 21.5%, and the mode purity is greater than 63.6%. However, the structure of multilayer substrates makes fabrication more complex. [27]. In conclusion, various attempts have been made to develop broadband vortex beam generators, but the mode purity bandwidth of the generator can be further improved.

There are two common approaches for the phase modulation of the metasurface. One is resonant modulation and the other is geometric (Pancharatnam-Berry) phase modulation. The former achieves the required phase distribution by adjusting the geometric size of the resonant meta-atoms [31], but the existence of dispersion in resonant atoms usually leads to a narrow bandwidth. The latter achieves phase control by rotating the meta-atom structure [32,33], but only for circularly polarized waves. A hybrid phase-tuning method has been used to achieve high-gain and broadband pencil beams by synthesizing the above two approaches [34].

In this paper, a hybrid phase-turning meta-atom that combines a circular ring patch and a windmill ring patch is proposed to broaden the mode purity bandwidth of vortex beams. A single-layer metasurface is designed to form the desired wavefront using the proposed meta-atom. The experimental results show that the proposed metasurface obtains a broadband vortex beam with mode *l* = −1 across 26.5–40 GHz. The generated vortex beam also features low crosstalk and wide 3 dB gain bandwidth. This means that our work can further promote the development of vortex beams in wireless communications.

## 2. Metasurface Design

Figure 1 illustrates the overall structure of the designed metasurface. The reflective metasurface is illuminated by a y-polarized wave and expected to transform a spherical wave into an OAM wave. Figure 2a shows the meta-atom structure, where the Taconic TLX-8 (ε_r_ = 2.55 and tanδ = 0.0019) is selected as the substrate and the metal patch is 0.017 mm thick copper. The parameters are given as follows: *r*_2_ = 0.44mm, *g* = 0.21, *h* = 1.14 mm, *w* = 0.1 mm, *P* = 4 mm [35]. The structure variation processes of meta-atoms with different *θ* are depicted in Figure 2b. Obviously, the dimension and orientation of the windmill ring patch vary simultaneously as the *θ* varies. Dual resonance can be generated to expand the phase range when the circular ring patch and the windmill ring patch reach the resonance size. This means that the proposed meta-atom is a hybrid phase-turning structure which uses both resonant and geometric phase modulation. Figure 3a reveals that the variation of the phase deflection is small within the entire range of 26.5 GHz to 40 GHz. Figure 3b depicts that the proposed meta-atom features low loss, which implies the proposed meta-atom exhibits broadband potential.

To obtain the required spiral phase distribution with the mode *l* = −1, the spherical wavefront excited by the feed needs to be converted into a spiral wavefront. As depicted in Figure 1, the feed illuminates the whole metasurface at oblique incidence. The vertical distance between the feed phase center and the top of the metasurface is 80 mm. To decrease the blocked effect of the primary feed, the angles of the incident beam and reflected beam are both adjusted to 15°. The phase difference between each meta-atom and the feed is compensated by changing the angle *θ*. The specific phase compensation equation is as follows:(1)$$\varphi_{mn}=l\cdot \varphi -k\times (d_{mn}-\vec{r_{0}}\times \vec{r_{mn}})$$
where *l* is defined as the mode of OAM (i.e., *l* = 0, ±, ±2…), *φ* represents the azimuthal angle around the center of the metasurface. *k* is a wavenumber in free space. *d*_mn_ represents the distance from the *mn*th meta-atom to the phase center of the feed. $\vec{r_{0}}$ is defined as the meta-atom vector of the reflected beam. The position vector of the *mn*th meta-atom is defined as $\vec{r_{mn}}$. The phase distribution of each meta-atom can be obtained through Equation (1), which is shown in Figure 4.

## 3. Experimental Result

To verify the performance of the generated vortex beam, a square metasurface composed of 400 meta-atoms is fabricated by using PCB technology, as shown in Figure 5a. Furthermore, detailed measurements are employed to evaluate the performance of the generated vortex beam. The work of the generator near-field measurement is carried out on the optical platform to sample the spatial field distribution, as shown in Figure 5b. The standard rectangular waveguide BJ320 (Beijing Xibao Electronic Technology Co., Ltd., Beijing, China) is chosen as the near-field receiving probe. The near-field receiving probe is 1150 mm from the metasurface. The observation plane size is 400mm × 400mm. The test frequency range is from 26.5 GHz to 40 GHz. The calculated results of the amplitude and phase distribution are extracted from the observation plane, as shown in Figure 6. It can be observed that typical annular amplitude distributions can be achieved across the test band. Furthermore, obvious anticlockwise helical phase distributions are observed, which indicates that the reflected beam is a vortex beam with the mode *l* = −1.

To calculate the purity of different OAM modes, the individual OAM modes are decomposed by Fourier transform. The corresponding equation is expressed as follows [36]:(2)$$A_{l}=\frac{1}{2\pi }\int_{0}^{2\pi }\psi (\varphi )e^{-jl\varphi }d\varphi$$
(3)$$\psi (\varphi )=\sum_{l}A_{l}e^{jl\varphi }$$
where *ψ(φ)* is defined as the field sampling function of the observation plane. The weight coefficients for different modes of OAM are calculated using Equation (4), and mode l is calculated from −4 to 4.
(4)$$\text{energy weight}=\frac{A_{l}}{\sum_{l\prime =-4}^{4}A_{l}\prime }$$

The OAM spectrum of Figure 6 shows that the energy weight of OAM mode *l* = −1 has occupied the most part of all modes at 35 GHz, and the simulated and measured mode purities are 83% and 80%, respectively. More importantly, the simulated and measured results indicate that the purity of the OAM dominant mode is greater than 70% across 26.5 GHz to 40 GHz, while the OAM mode purity bandwidth is about 38.57%. Measured values are lower than simulation results due to possible environmental noise and fabrication tolerances. However, high mode purity can be maintained in a wide frequency band as the mode *l* = −1, which corroborates the effectiveness of the proposed generator for widening the bandwidth of OAM mode purity.

Maximum crosstalk is taken into account to quantify the interference between different modes of OAM. The corresponding calculation Equation (5) is as follows:(5)$$\mathrm{crosstalkmax}=20\log10\frac{A_{1}}{A_{2}}$$
where *A_1_* and *A_2_* are the OAM modes with the first and second weight coefficients, respectively. The OAM beam propagation in the air generally has mode crosstalk due to signal path coupling, multipath effect, and imperfect transmitting and receiving devices. However, in the experimental environment, mode crosstalk is caused by discretization of the meta-atom phase aperture distribution [37]. The calculated values of the maximum crosstalk and dominant mode purity are listed in Table 1. The mode purity is greater than 71% and the maximum crosstalk is less than −14 dB, which means that the generated vortex beam can be suitable for more OAM-related applications, such as broadband vortex imaging.

To obtain the far-field radiation performance of the proposed generator, the far-field testing was performed in the anechoic chamber, which is shown in Figure 7. The generator is fixed on the metal rotating pedestal, and the rotating angle is set from −30° to +30°. The transmitting horn is 3 m from the metasurface. The received signal is transmitted into the spectrum analyzer to obtain the radiation patterns of the vortex beam. Figure 8 depicts the normalized radiation patterns of the generated vortex beam. It can be seen that the measured radiation patterns present a typical OAM beam with a peak appearance angle of about ±5°, and good directivity can be maintained in the wide band range. The results of the measurement and simulation in the main lobe region are in good agreement. The measured results show a higher value in the amplitude null compared to the simulated results, which is mainly caused by manufacturing errors, measurement errors, and alignment errors.

The gain spectrum and aperture efficiency of the generated vortex beam versus frequencies are plotted in Figure 9. The peak gain measured at 37 GHz is 22.3 dBi, and the aperture efficiency is 13.88%. The simulated gain variation is less than 3 dB over the range of 28.5–43.5 GHz (the relative bandwidth is 42.86%). However, the measured 3 dB gain range of the generated OAM beam is 28.5–40 GHz (the relative bandwidth is 32.85%), since the vector network analyzer has a maximum measurement frequency of 40 GHz. The above results reveal that the proposed OAM generator can simultaneously obtain wide OAM mode purity bandwidth and gain bandwidth.

The key performance indicators of the proposed OAM generator are compared with those of the previous works, which have been listed in Table 2. This work exhibits significant advantages in terms of mode purity bandwidth, crosstalk, and gain bandwidth. Moreover, the desired phase profile is achieved with only a single-layer dielectric substrate, which greatly reduces fabrication cost and design complexity. This demonstrates that the proposed generator is a good selection for vortex beam communication.

## 4. Conclusions

In summary, we have proposed a broadband OAM generator that can generate a vortex beam with the mode *l* = −1. The combination of resonant and geometric phase modulation methods effectively improves the mode purity bandwidth of the vortex beam generated via metasurface. The measured results demonstrate that the proposed generator has the merits of low crosstalk, wide mode purity bandwidth, high mode purity, and wide 3 dB gain bandwidth, which indicates that the generator is a promising solution for OAM-related applications in wireless communication systems.

## Figures and Tables

**Figure 1 micromachines-14-00465-f001:**
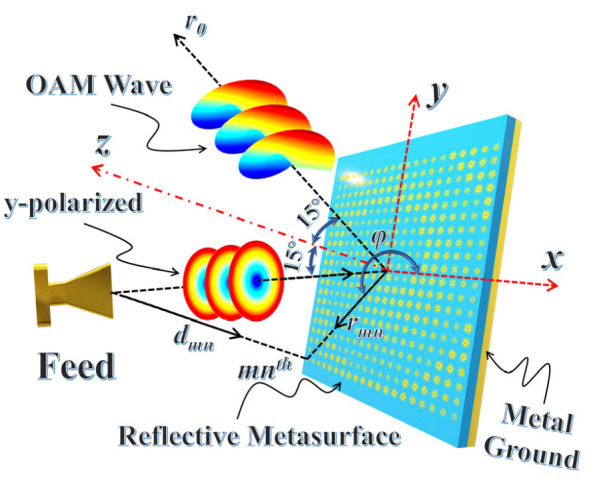
The theoretical model for generation and propagation of vortex beams.

**Figure 2 micromachines-14-00465-f002:**
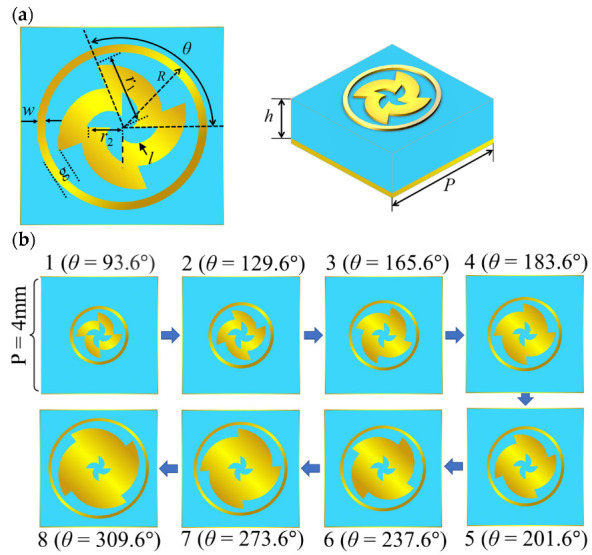
(**a**) Meta-atom structure; (**b**) schematic diagram of the variation of the meta-atom.

**Figure 3 micromachines-14-00465-f003:**
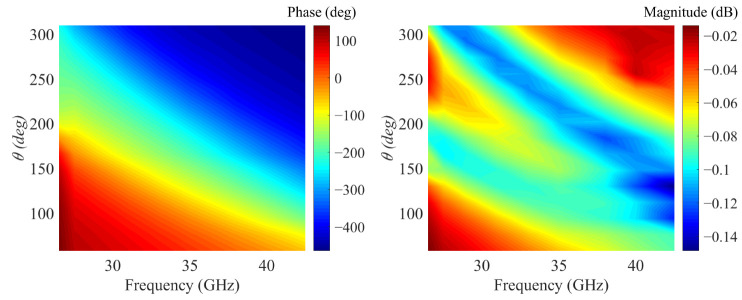
The simulated reflection phase and magnitude of the proposed meta-atom at different frequencies.

**Figure 4 micromachines-14-00465-f004:**
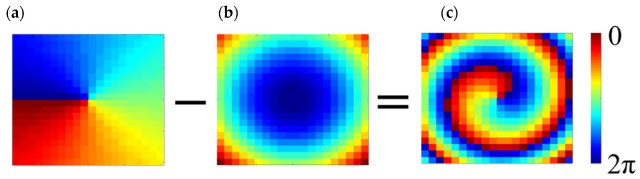
Phase compensation of the proposed metasurface: (**a**) the OAM desired phase, (**b**) the initial phase, (**c**) the final compensation phase.

**Figure 5 micromachines-14-00465-f005:**
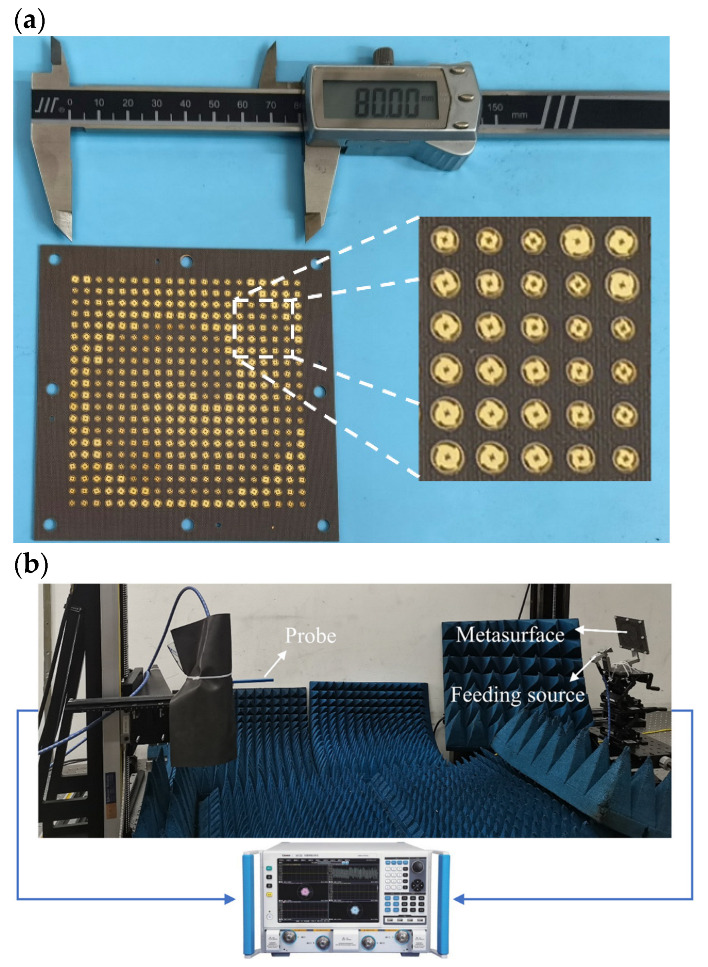
(**a**) The photograph of the fabricated metasurface; (**b**) near-field experimental system.

**Figure 6 micromachines-14-00465-f006:**
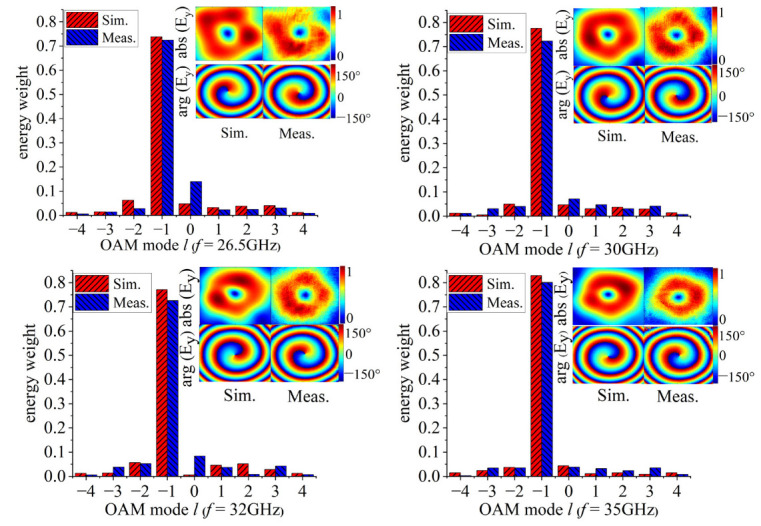
Simulated and measured phase and amplitude distributions in the observation plane and spectral calculation of OAM mode at different frequencies.

**Figure 7 micromachines-14-00465-f007:**
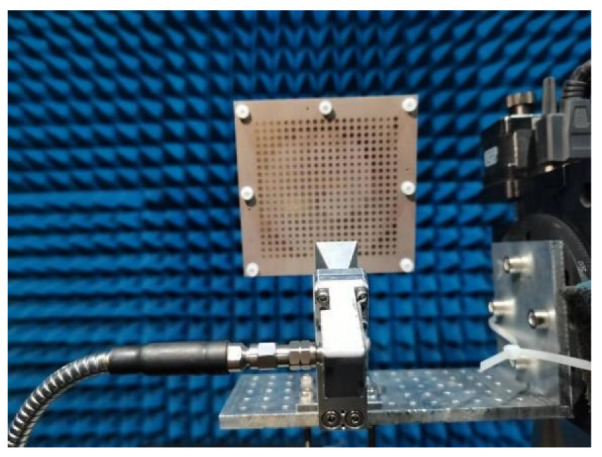
The prototype of OAM generator for generating vortex beams.

**Figure 8 micromachines-14-00465-f008:**
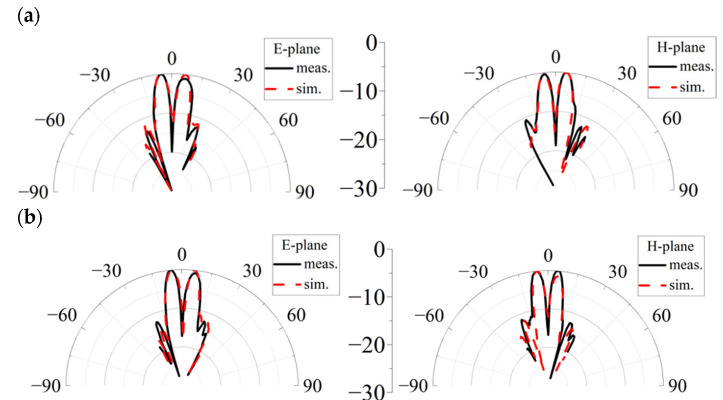
Simulated and measured normalized far-field patterns of the proposed OAM generator at (**a**) 28.5 GHz, (**b**) 30 GHz, (**c**) 35 GHz, (**d**) 40 GHz.

**Figure 9 micromachines-14-00465-f009:**
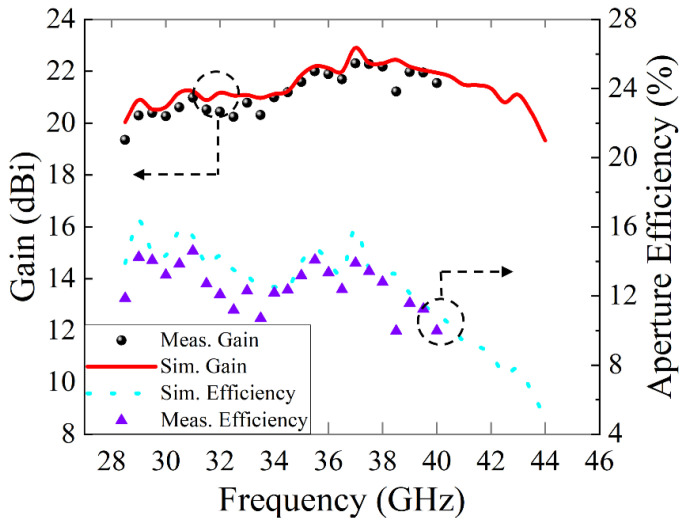
Gain and aperture efficiency of the generated vortex beam versus frequencies.

**Table 1 micromachines-14-00465-t001:** Purity and max crosstalk of measured vortex beam.

Frequency (GHz)	26.5	30	32	35	37	40
Measured Max. purity	72%	72%	73%	80%	76%	71%
Measured Max. Crosstalk (dB)	−14.28	−20.24	−18.76	−26.34	−18.02	−15.61

**Table 2 micromachines-14-00465-t002:** Comparison of the existing reflective OAM generator.

Ref.	Frequency (GHz)	Thickness	Aperture Size	No. of Substrates	Mode Purity Bandwidth	Peak AE.	Max. Crosstalk (dB)	3 dB Gain Bandwidth (%)
[25]	5.5	0.027λ_0_	2.57λ_0_	1	11% (>60%)	11.9%	—	—
[26]	30	0.6λ_0_	10λ_0_	1	21.7% (>60%)	8%	—	—
[27]	5.7	0.069λ_0_	2.38λ_0_	4	21.5% (>63.6%)	18.46%	—	—
[28]	67	0.14λ_0_	49λ_0_	1	17.1% (>75.5%)	—	<−10	—
[29]	12	0.1λ_0_	9.6λ_0_	2	22.5% (>75%)	20.2%	—	25%
This work	35	0.13λ_0_	9.3λ_0_	1	38.57% (>70%)	13.88%	<−14	42.86%

## Data Availability

The data supporting the findings of this study can be made available to genuine readers after contacting the corresponding authors.

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
