# Peer review of "A Broadband Vortex Beam Generator Based on Single-Layer Hybrid Phase-Turning Metasurface"

_micromachines, 2023, doi:10.3390/mi14020465_

Round 1

Reviewer 1 Report

This manuscript demonstrates the generation of OAM wave using metasurface in a wide frequency range. Both simulation and experiments have been shown. This manuscript in general is acceptable for publication when the following comments and suggestions are addressed.

1.      The introduction should include more typical work on the geometric phase based metsurfaces such as “IEEE Transactions on Antennas and Propagation, 65 (1), 396-400, 2017” and “IEEE Antennas and Wireless Propagation Letters, 18 (3), 477-481, 2019”.

2.      A shown in Fig. 1, there is a metal ground on the metasurface. Then why the reflection magnitude is not always 1 as in Fig.2 (b)?

3.      The author should mention the polarization of the metasurface.

4.      To make the results more reliable, at least the generation of a higher order OAM mode should be presented.

5.      Some figures and equations in the manuscript are very blurry. Please revise accordingly.  

Reviewer 2 Report

The authors proposed a broadband vortex generator, which has a good performance in terms of purity, bandwidth, and gain. I will recommend this manuscript to be published after addressing the follow issues:

1. please cite more existing broadband vortex metasurface such as: High-gain Broadband Millimeter-wave Multidimensional Metasurface for Generating Two Independent Vortex Waves.

2. The authors claim they used the hybrid phase turning metasurface, PB phase, for linearly polarized incidence. However, as I know that the PB phase is used in circularly polarized, please clarify this in the manuscript.

Reviewer 4 Report

Dear Authors and Editors,

The manuscript details the design and characterization of an OAM generating metasurface, with the main novelty over previous demonstrations being the increased bandwidth, which indeed is an important aspect for metasurfaces given their resonance-dependent behaviour. In my opinion it is an interesting and relevant piece of work. However, I think there are a couple of points that need to be addressed, some to clearly explain the methodology and some that would add value for the reader and hopefully increase the impact of the paper.

1.       The meta-atom structures have an interesting shape. How did the authors arrive at this shape, and what is the logic behind it?

2.       In the abstract it is stated that the incident illumination is linearly polarized, but I did not find any more information on this in the manuscript. When incident on the metasurface was it TE, TM or some combination? (This could also be indicated in Figure 1a). And how was the choice made?

3.       What determined the choice of incidence angle? And are there any limitations or trade-offs if increasing this angle?

4.       In line 89, a “blocked effect” is mentioned, is this a change in beam profile upon reflection?

5.       In Figure 2 the vertical axis shows “theta”, am I correct in assuming this is when changing both the angle (“theta”) and the size of the meta-atoms?

6.       The angles of the meta-atoms, “theta” are clearly labelled in Figure 1c, however, the size of the meta-atoms is not apparent. A scale bar should be added, and the size of the unit cell should be mentioned in the text.

7.       The reading on the calipers in Figure 4 is hard to read, especially on paper, consider removing it and adding a scale bar instead.

8.       The manuscript only considers OAM of order 1, have the authors looked into higher order OAM states? Are there any additional challenges to solve if one wants to do this?

9.       How were the simulations done – what method/software? And what material parameters were used? What is the thickness of the different layers?

10.   How was the fabrication done? And with what materials?

Round 2

Reviewer 4 Report

Thank you for answering my questions and comments.

The manuscript has been improved, however, I still feel it lacks some of the details the authors mentioned in the cover-letter. Such as the materials used and a comment about the simulations. Also, some of the figure cations are very minimalistic, for example in Figure 3 it should be mentioned that this is simulation results.

With best regards,
